# Cetuximab as a Key Partner in Personalized Targeted Therapy for Metastatic Colorectal Cancer

**DOI:** 10.3390/cancers16020412

**Published:** 2024-01-18

**Authors:** Nadia Saoudi González, Javier Ros, Iosune Baraibar, Francesc Salvà, Marta Rodríguez-Castells, Adriana Alcaraz, Ariadna García, Josep Tabernero, Elena Élez

**Affiliations:** 1Vall d’Hebron Institute of Oncology (VHIO), 08035 Barcelona, Spain; nsaoudi@vhio.net (N.S.G.); fsalva@vhio.net (F.S.);; 2Vall d’Hebron Hospital Campus, 08035 Barcelona, Spain

**Keywords:** cetuximab, colorectal cancer, *KRAS*G12C, *BRAF*V600, personalized treatment

## Abstract

**Simple Summary:**

The treatment of colorectal cancer has been revolutionized in recent decades by the identification of new targeted agents, particularly for metastatic colorectal cancer when the disease has spread to other sites in the body. Cetuximab targets a specific protein on the surface of the tumor cells. This review highlights the expanding role of this drug in diverse types of metastatic colorectal cancer and describes how cetuximab can be partnered with other anticancer drugs such as immunotherapy (a therapy that activates the immune system) and other agents targeting cell proteins. We also describe promising innovative ongoing clinical trials. The evolving importance of cetuximab in anticancer therapy offers renewed hope in the landscape of personalized treatment for patients with metastatic colorectal cancer.

**Abstract:**

Cetuximab, a chimeric IgG1 monoclonal antibody targeting the epidermal growth factor receptor (EGFR), has revolutionized personalized treatment of metastatic colorectal cancer (mCRC) patients. This review highlights the mechanism of action, characteristics, and optimal indications for cetuximab in mCRC. Cetuximab has emerged as a pivotal partner for novel therapies in specific molecular subgroups, including *BRAF* V600E, *KRAS* G12C, and HER2-altered mCRC. Combining cetuximab with immunotherapy and other targeted agents further expands the therapeutic landscape, offering renewed hope for mCRC patients who face the development of resistance to conventional therapies. Ongoing clinical trials have continued to uncover innovative cetuximab-based treatment strategies, promising a brighter future for mCRC patients. This review provides a comprehensive overview of cetuximab’s role and its evolving importance in personalized targeted therapy of mCRC patients, offering valuable insights into the evolving landscape of colorectal cancer treatment.

## 1. Introduction

Colorectal cancer (CRC) ranks as the third most prevalent cancer worldwide and is the second leading cause of cancer-related deaths [1]. Over the past few decades, significant strides have been made to improve the survival outcomes of individuals diagnosed with metastatic CRC (mCRC); this includes the introduction of novel and more effective therapies, notably immunotherapy, targeted therapies for specific molecularly selected populations, and monoclonal antibodies (mAb). In addition to these therapeutic advances, there has been notable progress in the management of localized metastases and early detection techniques for CRC. Taken together, these strategies have increased the median overall survival (OS) of patients diagnosed with mCRC from 20 months to around 30 months [2]. However, the primary treatment approach for the majority of patients suffering from mCRC continues to revolve around cytotoxic agents based on 5-fluorouracil (5-FU) combined with leucovorin and irinotecan or oxaliplatin, which may be combined with biological agents in the first and second lines, depending on the patient’s *KRAS* status.

In patients with mCRC who harbor activating mutations of the mitogen-activated protein kinase (MAPK) pathway, research has shown the importance of recognizing that targeting a specific molecular alteration in this pathway can elicit different responses in patients with different tumor histologies; remarkably different outcomes were reported for patients with mCRC harboring *BRAF* V600 mutations compared with those with metastatic melanoma, and likewise for patients with mCRC harboring a *KRAS* G12C mutation compared with metastatic lung cancer [3,4]. This phenomenon was subsequently explained by studies revealing that in mCRC, inhibiting signaling at one point of the MAPK pathway can activate the pathway through positive feedback loops at other action points, highlighting the significance of tissue specificity [5,6]. As a result, targeted monotherapy treatments against activating mutations of the MAPK pathway in mCRC have demonstrated limited activity [3,7]. For instance, while the exclusive inhibition of BRAF in patients with *BRAF* V600 mCRC or *KRAS* G12C mutations has not shown significant efficacy [4], combining a second point of inhibition, such as the upstream epidermal growth factor receptor (EGFR), substantially improved the overall response rates (ORRs) and enhanced the effectiveness of targeted therapies. On the other hand, individuals with microsatellite instability (MSI) in mCRC can experience significant benefits from first-line immunotherapy [8,9].

Cetuximab, the first monoclonal antibody (IgG1) targeting EGFR, and the subsequently developed panitumumab, which also targets EGFR, have shown improved outcomes either alone (in the refractory setting) or combined with chemotherapy as effective treatments, with the associated characteristic skin toxicity. However, *RAS* mutations are linked to anti-EGFR resistance, necessitating an assessment of the mutational status of *RAS* prior to treatment [2]. In various settings, first-line chemotherapy in combination with cetuximab or panitumumab demonstrated improved progression-free survival (PFS), OS, and ORRs in *RAS* wild-type (wt) mCRC patients [10,11,12,13,14,15,16,17,18,19]. The tumor’s location plays an important role in treatment decisions in mCRC, with left-sided tumors showing significant benefits from anti-EGFR mAbs, whereas right-sided tumors may not derive the same benefits in terms of PFS and OS [20].

Combination therapies using *BRAF* inhibitors and drugs that block them at different points within or upstream of the MAPK pathway, specifically MEK and EGFR inhibitors, have become standard cancer treatments for patients with BRAFV600 mCRC since the discovery of an adaptive feedback reactivation loop of MAPK signaling that sustains the activation of MAPK upon the inhibition of *BRAF*.

The example of suppressing *BRAF* illustrates the importance of the inhibition of EGFR for precision medicine in patients with mCRC harboring specific molecular alterations. In this review, we summarize published evidence highlighting the significant role of cetuximab as a key factor in targeted therapy for patients with mCRC harboring targetable molecular alterations. Figure 1 shows the MAPK pathway, delineating the key inhibitory sites synergistically targeted by cetuximab for a comprehensive therapeutic intervention.

## 2. Mechanism of Action of Cetuximab and Panitumumab, and the Current Treatment Paradigm in mCRC

Cetuximab, the first EGFR-targeting monoclonal antibody approved for mCRC, is a chimeric IgG1 antibody. It specifically binds to the EGFR Domain III with higher affinity than natural ligands such as EGF and TGF-α, effectively blocking ligand-induced receptor activation. Cetuximab also aids in downregulating EGFR-dependent signaling by promoting the internalization of EGFR [21]. Panitumumab, a fully human IgG2 monoclonal antibody, binds to EGFR with higher affinity than cetuximab, inhibiting the activation of EGFR without inducing activation of the immune system [22]. Cetuximab is associated with more frequent infusion reactions due to its chimeric nature. Pharmacokinetic differences include cetuximab’s non-linear clearance and longer half-life, while panitumumab exhibits both linear and non-linear clearance mechanisms and a shorter half-life [23].

Early Phase I trials focused on determining the cetuximab dosages associated with saturation of clearance, indicative of complete EGFR occupancy. These studies revealed that clearance from the bloodstream was relatively slow, with a median half-life of 7 days [24]. Cetuximab undergoes lysosomal degradation through the reticuloendothelial system and protein catabolism via a target-mediated drug disposition pathway.

In the clinical management of mCRC patients, it is crucial to determine the presence of *RAS* mutations up front, as these are indicators predicting a poor response to the inhibition of EGFR [15,17,18]. Several clinical trials with cetuximab and panitumumab have shown that not only mutations in a specific part of the *KRAS* gene (Exon 2) but also mutations in other regions of *KRAS* (Exons 3 and 4), as well as Exons 2, 3, and 4 of the *NRAS* gene (expanded *RAS* analysis), are linked to a lack of effectiveness of anti-EGFR monoclonal antibodies [15,18]. On the basis of this, the use of EGFR inhibitors as a first-line treatment is restricted to patients with *RAS*wt mCRC.

Retrospective analysis of the pivotal clinical trials in mCRC further demonstrated that the benefit EGFR inhibitors as the first line depends on the tumor’s location within the colon [2,25]. Patients with left-sided tumors benefit greatly from anti-EGFR mAbs, with a higher ORR and a significantly longer PFS and OS [20]. However, in right-sided tumors, the addition of cetuximab or panitumumab to chemotherapy does not improve PFS and OS, although the ORRs are higher. Therefore, the preferred approach for right-sided tumors, regardless of the mutation status of *RAS*, is to combine doublet or triplet chemotherapy with the antivascular endothelial growth factor (VEGF) antibody bevacizumab instead [2]. Anti-EGFR mAbs are considered only for right-sided *RAS*wt tumors when a strong response is needed, such as for conversion therapy. In cases of frail or elderly patients who cannot tolerate chemotherapy but have left-sided *RAS*wt tumors, anti-EGFR mAbs as a monotherapy may be an alternative [10]. It is important to note that combining anti-EGFR mAbs with chemotherapy is not advised for tumors with *BRAF* mutations.

The recent PARADIGM trial was the first study to compare the effectiveness of an anti-EGFR antibody with an anti-VEGF antibody when combined with the standard fluorouracil plus oxaliplatin chemotherapy regimen in *RAS*wt left-sided mCRC [26]. The study showed that patients who received first-line treatment with mFOLFOX6 and panitumumab had a longer OS compared with those treated with bevacizumab and FOLFOX. EGFR inhibitors achieved a benefit in the second line when combined with irinotecan or FOLFIRI, albeit with no benefit in terms of OS [27,28]. In the third-line setting, cetuximab and panitumumab have shown efficacy as single agents in mCRC after standard chemotherapy. Cetuximab improved OS and PFS compared with best supportive care (BSC), while maintaining quality of life [29]. Panitumumab also increased PFS and ORR compared with BSC in mCRC patients previously treated with standard chemotherapy (with no difference in the OS due to crossover) [30]. When comparing cetuximab with panitumumab in the refractory setting, no statistically significant difference was found in OS, and the toxicity profiles were similar [30]. In *RAS*wt patients who had not received prior anti-EGFR monoclonal antibodies, a combination of irinotecan and cetuximab in the third or later lines of treatment was more effective than cetuximab alone in terms of the ORR and PFS, though not the OS [31]. Thus, the combination of irinotecan and cetuximab can be considered for this patient population and clinical setting.

Circulating tumor DNA (ctDNA) analysis offers valuable insights into the dynamics of mCRC treatment. Initial ctDNA studies revealed the expansion of subclonal *KRAS* mutant clones during anti-EGFR therapy as sensitive cells dwindled, leading to progression of the disease [32,33]. Subsequent non-EGFR treatments may partially restore sensitivity, setting the stage for anti-EGFR rechallenge. The decay of resistant clones during non-EGFR-based treatment revealed a half-life of around 3.7 to 4.7 months, indicating ongoing clonal evolution during therapy [33]. The CRICKET trial, initially a retrospective ctDNA analysis, led to the prospective CHRONOS study, utilizing ctDNA to select candidates for rechallenge with cetuximab and irinotecan in mCRC. In the CRICKET trial, 28 *RAS* and *BRAF* wild-type mCRC patients who were resistant to the initial cetuximab-based therapy showed a 21% overall response rate (ORR) and positive disease control [34]. Baseline ctDNA analysis correlated the RAS mutations with shorter progression-free survival times, guiding patient selection. Innovative ctDNA-guided strategies, exemplified by the CHRONOS trial, demonstrated promising results in selecting refractory mCRC patients for anti-EGFR rechallenge, with a 63% disease control rate [35]. However, further validation through larger randomized trials is essential to solidify its role in clinical practice.

## 3. Partnering Cetuximab with Targeted Therapy in mCRC

### 3.1. BRAF V600E Inhibition and Cetuximab

Approximately 8 to 12% of patients with mCRC have a *BRAF* V600E mutation, which causes constitutive activation of this kinase and results in a shorter OS and a lower ORR with chemotherapy [36,37,38]. The addition of anti-VEGF therapy to chemotherapy improved clinical outcomes in *BRAF* V600E mCRC patients, while the addition of an anti-EGFR to a chemotherapy backbone is not recommended with this mutation [2,39,40,41]. In contrast to the excellent outcomes in melanoma, single-agent *BRAF* inhibitors showed little to no clinical activity in patients with *BRAF* V600E-mutated mCRC [42,43].

Preclinical investigations to understand the variations in the histological responses to monotherapy *BRAF* inhibitors uncovered a resistance mechanism involving the activation of EGFR through feedback when BRAF inhibitors are present in the context of mCRC [5,44]. As a result, different combinations of BRAF and MEK inhibitors, anti-EGFR treatments, and chemotherapy have been investigated in clinical trials involving patients with *BRAF* V600 mCRC [42,45,46,47]. Nevertheless, the majority of these combinations have demonstrated restricted clinical effectiveness, yielding only slight enhancements in the ORR and median PFS.

The BEACON trial illustrated that the combination of *BRAF* inhibition and anti-EGFR therapy using cetuximab yielded better results compared with irinotecan-based chemotherapy in refractory *BRAF* V600E mCRC patients. The trial evaluated encorafenib plus cetuximab, with or without the MEK inhibitor binimetinib (i.e., doublet or triplet therapy), versus irinotecan-based chemotherapy plus cetuximab. The median OS was 9.3 months for both the triplet and doublet therapies, whereas it was 5.9 months for the control group [8]. The MEK inhibitor’s role in this population remains to be defined. Recent data have suggested that patients with a high *BRAF* allele fraction in their ctDNA were associated with a more aggressive disease and poorer outcomes, demonstrated by the worse PFS and OS. Furthermore, patients with a high *BRAF* allelic fraction appeared to benefit more from triplet therapy, suggesting its potential as a marker for stratifying treatment [48].

The adverse effects linked to encorafenib combined with cetuximab were generally of mild to moderate severity, were manageable with appropriate care, and rarely led to discontinuation of the treatment. Changes to the treatment were seldom required [8,49,50]. Notable adverse effects in the BEACON study with this combination encompassed gastrointestinal toxicities, skin reactions, joint pain, muscle discomfort, kidney-related toxicity, fatigue, weakness, headaches, and fever. Skin malignancies were infrequent and resolved. These adverse effects are in line with what is expected on the basis of the known profiles of cetuximab and encorafenib, as well as their mechanisms of action.

Studies have shown that upon the development of resistance mechanisms with *BRAF* inhibitor combinations, various genetic alterations including *MAP2K1*, *GNAS*, *ARAF*, *PTEN*, *ERBB2*, *MEK*, and *EGFR* mutations, as well as the amplification of *KRAS*, *MET*, and *EGFR*, have been observed in both plasma and tissue samples [45]. Data from ctDNA analysis in the BEACON trial were recently disclosed. The prevalent changes in acquired resistance were *KRAS* and *NRAS* mutations, along with the amplification of *MET*, detected in both doublet and triplet treatment groups. In contrast, the control group displayed notably fewer instances of these alterations (0% in the controls versus 40% in the target therapy group). These changes at the end of treatment appeared to be subclonal, and many patients who encountered these mutations displayed multiple alterations simultaneously [51]. These data confirmed the importance of selective environmental pressure with targeted treatment.

The ANCHOR study, a single-arm Phase II study evaluating the encorafenib/cetuximab/binimetinib triplet in patients with treatment-naive *BRAF* V600E mCRC, was developed as a result of the encouraging preliminary findings of the safety run-in phase of the BEACON trial; the ORR was 48% with a median follow-up of 14.4 months, and the median PFS and median OS were 5.8 months and 17.2 months, respectively [52]. Despite this clinical activity, standard chemotherapy with a fluoropyrimidine/oxaliplatin doublet and an anti-VEGF agent has shown even better results [53], indicating that the *BRAF* inhibition combination may not be sufficient for *BRAF*-mutant mCRC as an initial treatment.

The Phase III randomized BREAKWATER clinical trial is currently assessing the optimal first-line treatment approach for *BRAF* V600E microsatellite stable (MSS) mCRC patients. The three treatment options in this trial are encorafenib–cetuximab, encorafenib–cetuximab–FOLFOX, and conventional chemotherapy (the investigator’s choice). Two cohorts of patients who had only one prior line of treatment and who received either encorafenib–cetuximab–FOLFIRI or encorafenib–cetuximab–FOLFOX were treated during the safety run-in phase. Strong responses and significant clinical activity were seen in the preliminary findings from the safety run-in (an ORR of 68% and 67%, and a median PFS of 9.9 months and non-estimable in the first-line treatment with encorafenib–cetuximab–FOLFOX and encorafenib–cetuximab–FOLFIRI, respectively) [54].

*BRAF* V600E/MSI tumors represent up to 30% of *BRAF* tumors [9]. The KEYNOTE-177 trial explored pembrolizumab as a first-line treatment option for MSI mCRC patients, comparing it with conventional chemotherapy [9]. When it came to patients with tumors harboring both a *BRAF* V600E mutation and MSI, the PFS was notably improved when patients were treated with the anti-programmed death-1 (PD-1) drug as the initial therapy.

Translational studies have suggested that the *BRAF*–MAPK signaling pathway creates an immunosuppressive environment, and inhibiting *BRAF* can enhance tumors’ antigen presentation, T-cell activation, and expression of PD-L1 [55]. Combining *BRAF* inhibitors with immune checkpoint blockade has been supported by preclinical data.

In a recent Phase II study, the combination of a *BRAF* inhibitor (dabrafenib), a MEK inhibitor (trametinib), and a PD-1 inhibitor (spartalizumab) in *BRAF* V600E-mutant mCRC patients proved to be well tolerated and showed an ORR of 25% [56]. Combining encorafenib and cetuximab with pembrolizumab in MSI-high (MSI-H)/dMMR and *BRAF* V600E-mutant mCRC patients was founded on the concept that tumor progression in this specific subgroup is driven by both genomic instability (reflected by MSI-H/dMMR) and *BRAF*-mutation-induced MAPK signaling, which are potential therapeutic targets. In the ongoing Phase II SEAMARK trial (NCT05217446), patients with previously untreated *BRAF* V600E and MSI-positive tumors are randomly allocated to receive either encorafenib, cetuximab, and pembrolizumab or pembrolizumab alone as part of the investigational treatment.

### 3.2. Inhibition of KRAS G12C and Cetuximab

*RAS* mutations play a significant role in various cancer types, driving oncogenic processes by disrupting the activity of GTPase, leading to uncontrolled activation of the downstream signaling pathway, including RAF proteins and MAPK [57]. Mutations in *KRAS*, particularly in Exon 2’s Codons 12 and 13, are found in 40% of mCRC cases and are associated with a worse prognosis [58,59]. Historically, targeting abnormal KRAS proteins has been considered challenging due to their small size, limited binding sites, and their tightly bound GTP in the active form. Notably, these mutations have always been negative biomarkers of a response to anti-EGFR in mCRC.

*KRAS* G12C mutations occur in 3% to 4% of mCRC patients, driving the tumor’s growth by suppressing the hydrolysis of GTP, which causes *KRAS* to transition to an active GTP-binding state and activate pro-tumorigenic effector signaling [60].

Recent developments have introduced small molecules targeting *KRAS* G12C. Sotorasib (AMG 510), a first-in-class targeted therapy, uses a unique mechanism by selectively and irreversibly inhibiting the kinase activity of *KRAS* G12C, effectively trapping it in the inactive GDP-bound state. In a Phase I clinical trial encompassing various solid tumors, sotorasib displayed a modest 7% response rate in 42 *KRAS* G12C mCRC patients, contrasting with the higher response rates (up to 32%) observed in other cancer types such as non-small cell lung cancer (NSCLC) [3]. Further translational studies investigating these discrepancies among histologies revealed that the predominant resistance mechanism in mCRC models for *KRAS* G12C inhibitors is the reactivation of EGFR signaling, in contrast to NSCLC. Consequently, subsequent clinical trial designs involving *KRAS* G12C inhibitors have incorporated EGFR inhibition on the basis of this compelling evidence. Sotorasib was combined with panitumumab in 40 chemorefractory *KRAS* G12C mCRC patients in the CodeBreak 101 study. The confirmed 30% ORR of the combination was threefold higher than that previously reported for sotorasib monotherapy, with a disease control rate of 93% [61]. Recently, the Phase III Codebreak 300 data have been published [62]. In this Phase III trial, mCRC patients who had not received previous treatment with a *KRAS* G12C inhibitor were randomized to receive sotorasib at a dose of 960 mg once daily plus panitumumab (53 patients), sotorasib at a dose of 240 mg once daily plus panitumumab (53 patients), or the investigator’s choice of trifluridine–tipiracil or regorafenib (standard care; 54 patients). After a median follow-up period of 7.8 months, both the 960 mg and 240 mg sotorasib–panitumumab groups exhibited longer median progression-free survival times (5.6 months and 3.9 months, respectively) compared with the standard-care group (2.2 months). These findings highlight the potential of this dual inhibitor approach in addressing KRAS G12C-mutated colorectal cancer.

Other *KRAS* G12C inhibitors have been developed. These include adagrasib (MRTX849), which has shown promise in various tumor types, including CRC [63]. However, adaptive feedback through the activation of EGFR (such with sotorasib) may limit its efficacy. The combination of adagrasib with cetuximab demonstrated clinical activity in heavily pretreated mCRC patients, with an ORR of 46% and a median response duration of 7.6 months [64].

Divarasib (GDC-6036) is a potent and highly selective covalent inhibitor specifically designed to target *KRAS* G12C. It functions by binding to the cysteine residue within the protein, permanently locking it into an inactive state, thereby halting its oncogenic signaling. In vitro studies have demonstrated that divarasib is significantly more potent, between 5 to 20 times more potent, and up to 50 times as selective as sotorasib and adagrasib [65]. In a Phase I study, divarasib was evaluated in 137 patients with advanced or metastatic solid tumors harboring the *KRAS* G12C mutation. In CRC patients (55 patients), a confirmed response was observed in 29% of patients, with a median PFS of 5.6 months [66]. In an ongoing Phase I study (NCT04449874), 29 patients with mCRC *KRAS* G12C were treated with divarasib and cetuximab. A substantial 66% of patients achieved a partial response, with a confirmed ORR of 62%, while the safety profile was manageable [67].

Table 1 presents published results of trials evaluating *KRAS* G12C inhibitors in combination with cetuximab or other EGFR inhibitors, and the main ongoing clinical trials of these in combination with EGFR inhibitors.

### 3.3. HER2 and Cetuximab

The prognostic role of the amplification of HER2 remains uncertain, but its significance as a predictor of resistance to EGFR inhibitors has been well established by preclinical studies and retrospective clinical data [68,69,70]. Persistent activation of the ERK1/2 pathway, driven by the amplification of *HER2*, contributes to anti-EGFR resistance. Studies have indicated that abnormal HER2 signaling, stemming from the amplification of HER3-activating ligands, counteracts cetuximab’s growth inhibition via the activation of ERK1/2 [71]. Xenografts derived from mCRC patients were used to explore the molecular changes linked to inherent resistance against anti-EGFR treatments. Mutations in *KRAS*, *NRAS*, and *BRAF* and elevated expression of HER2 correlated with the absence of positive outcomes in response to cetuximab [72].

In the clinical setting, the negative predictive value of the amplification of HER2 for anti-EGFR therapies was demonstrated through retrospective analyses. Patients with HER2 amplification receiving anti-EGFR therapy experienced notably shorter PFS compared with those without amplification [73]. However, not all published data have supported the negative predictive role of HER2-positivity in relation to anti-EGFR agents [74,75]. Prospective data are needed to accurately define the relationship between HER2 and anti-EGFR responsiveness.

Targeted therapy is being explored through various strategies, with diverse trials evaluating the role of HER2 as a positive predictive biomarker. The HERACLES trial demonstrated success with trastuzumab plus lapatinib, achieving a 30% response rate [76]. Pertuzumab plus trastuzumab in the MyPathway study achieved a 38% response rate in refractory mCRC patients with molecularly altered HER2 [77]. Notably, the MOUNTAINEER trial reported an impressive 55% ORR using tucatinib and trastuzumab in patients with HER2-amplified CRC [78]. In the HER2 landscape, the DESTINY-CRC01 study evaluating trastuzumab–deruxtecan in HER2-positive refractory mCRC proved pivotal, resulting in a 45% ORR with a median PFS of 4.1 months, and the median OS was not reached [79].

Neratinib, an irreversible pan-EGFR receptor tyrosine kinase inhibitor, demonstrated greater potency than lapatinib across various contexts, including cell lines and xenografts [80,81]. Given the significance of EGFR and its inhibition in *KRAS*wt CRC, a team of researchers hypothesized that the combined blockade of EGFR and other HER2 receptors might overcome resistance to cetuximab. Consequently, they formulated a Phase Ib trial to explore this amalgamation within the quadruple-WT (*KRAS*, *NRAS*, PIK3CA, and *BRAF*) subgroup of patients who had previously received cetuximab treatment. The inclusion of amplified HER2 was not mandatory for participation in the trial. Among the 16 patients who had biopsy data available after receiving anti-EGFR therapy upon enrollment, four (25%) exhibited the amplification of HER2. Despite the neratinib–cetuximab combination proving to be well tolerated, none of the patients achieved the necessary tumor shrinkage per RECIST 1.1, which is required for calculating the ORR [82]. Preclinical models have underscored the potency of the dual EGFR–HER-family blockade in cetuximab-resistant patient-derived xenograft models, emphasizing the possible efficacy of targeting the upstream pathways. An ongoing Phase II trial is presently investigating the effectiveness of two treatment protocols: one involving neratinib and trastuzumab, and the other involving neratinib and cetuximab (NCT03457896). This investigation is focused on patients with mCRC harboring the “quadruple wild-type” genetic profile. Patient stratification within the study hinges on their HER2 status, which can be categorized as amplified, non-amplified (wild-type), or mutated.

HER2 mutations introduced into colorectal cancer cells demonstrated resistance to cetuximab and panitumumab. Furthermore, the sequencing of 48 cetuximab-resistant patient-derived xenografts (PDX) identified HER2 mutations. While treatment with a single HER2-targeted drug delayed the growth of tumors, dual HER2-targeted therapy with trastuzumab plus tyrosine kinase inhibitors, such as neratinib, produced a regression of these HER2-mutated PDX [83]. These results have established a robust preclinical foundation, suggesting that HER2-activating mutations may serve as a potential target for the treatment of colorectal cancer.

## 4. Immunotherapy and Cetuximab

The mechanism of action of cetuximab involves, in addition to the inhibition of the EGFR–RAS–MAPK pathway, the induction of antigen-dependent cell cytotoxicity (ADCC). Patients receiving IgG1-based monoclonal antibodies undergo ADCC. In contrast, treatment with other isotypes, such as IgG2 mAbs, lacks this immune-mediated response of the host [84]. Given this context and the recent advancements in immune checkpoint inhibitors (ICI) for cancer therapy, a potentially effective strategy involves co-administering IgG1-isotype monoclonal antibodies, such as cetuximab. This approach aims to enhance the antitumor efficacy of ICIs. Besides fostering functional communication between the dendritic cells and natural killer (NK) cells, cetuximab treatment has the potential to recruit cytotoxic T cells to the tumor microenvironment [85,86]. This recruitment could prime the immune system for a more favorable response to ICI treatment. In retrospective studies performed in the mCRC tumor microenvironment, the introduction of anti-EGFR mAbs with concomitant chemotherapy was linked with higher quantities of cytotoxic CD8+ cells, memory-effector CD45RO+ cells, and regulatory FOXP3+ T cells within metastatic sites compared with scenarios without treatment, with chemotherapy alone, or chemotherapy coupled with antiangiogenic mAbs [87]. Additionally, the combination of anti-EGFR treatment with chemotherapy resulted in increased occurrences of cells presenting the inhibitory immune checkpoint molecule PD-1. Activation of the NK cells by cetuximab induced the expression of PD-1. On the contrary, inhibition of PD-1 not only boosted the activation of cetuximab-induced NK cells, making them more potent against tumor cells expressing PD-Ligand 1 (PD-L1), but in patients with advanced CRC, cetuximab treatment also amplified intratumoral cytotoxic T lymphocytes (CTLs) and the presence of inhibitory immune checkpoint molecules such as those blocking the PD-1/PD-L1 pathway [88,89]. On the other hand, strategies evaluating anti-PD-1/PD-L1 monotherapy for mCRC MSS patients failed to show activity [90]. In this context, different combinations of cetuximab with immunotherapy have been evaluated and are ongoing to test the hypothesis that combining the inhibition of PD-1 with an anti-EGFR antibody can induce a more favorable immune tumor microenvironment and overcome PD-1 resistance; however, to date, these combinations have demonstrated limited activity. Table 2 includes the main ongoing and published clinical trials investigating cetuximab combined with immunotherapy.

In the Phase II CAVE study, patients with *RAS*wt mCRC who responded to first-line chemotherapy with an anti-EGFR mAb and then received second-line systemic therapy were given a combination of cetuximab and the PD-L1 mAb avelumab in the third or later lines of therapy as a rechallenge strategy. The ORR was 7%, with a median OS of 11.6 months (95% CI, 8.4–14.8), and the median PFS was 3.6 months (95% CI, 3.2–4.1) [91]. This ongoing study will compare this combination with cetuximab alone (NCT05291156).

In another study of MSS mCRC, the combination of chemotherapy, cetuximab, and the anti-PD-L1 antibody avelumab was explored. Among 43 MSS RAS/*BRAF* wild-type mCRC patients, the trial identified PD-L1 mutations in high-affinity FcγR3a-expressing patients. These mutations reduced PD-L1 levels via the decay of RNA and protein degradation, hampering the binding of avelumab but enhancing the killing of mutant cells by T cells, and therefore mediating the subclonal immune escape from avelumab [92].

An interim analysis of the AVETUXIRI trial (NCT03608046), a Phase I study, has recently been conducted. This trial investigated the combination of cetuximab, irinotecan, and avelumab. The triple regimen demonstrated a manageable safety profile, with the most significant Grade 3–4 adverse event being diarrhea, which was resolved through adjustments to the dose. Notably, among patients with *RAS*wt tumors, 3 of 10 patients achieved a response (ORR of 30%), with 6-month PFS and 12-month OS rates of 40.0% and 50.0%, respectively. With these promising initial findings, the study is continuing, with more comprehensive results expected as the trial progresses [93].

The AVETUX Phase II trial explored first-line therapy for MSS mCRC patients by combining standard CT with FOLFOX, cetuximab, and avelumab (NCT03174405). This study enrolled 43 patients, achieving an ORR of 79.5%, including 6 complete responses and 25 PRs [94]. These results warrant further exploration in a randomized trial. Notably, translational analyses demonstrated an early reduction in ctDNA levels, signifying a rapid tumor response, while the diversity of immune cells and tumor-infiltrating lymphocytes after three treatment cycles correlated with improved disease outcomes [95].

Although having a partial local immunologic impact on tissue specimens, the combination of cetuximab and pembrolizumab showed no activity in patients with *RAS*wt mCRC. This combination was tested in patients with refractory mCRC *RAS*wt in a Phase Ib/II study. Changes in the tumor microenvironment were analyzed at the baseline and treated tumor tissue using flow cytometry and multispectral immunofluorescence. In 44 patients evaluated for effectiveness, the primary goal was not achieved (2.6% ORR, 31% 6-month PFS). No notable increase in adverse events was seen. After treatment, positive immune modulation was seen, with a 47% increase in intratumoral cytotoxic T lymphocytes, this effect being more noticeable in patients experiencing tumor shrinkage (*p* = 0.05) [96].

Ongoing research is exploring new approaches that combine cetuximab with immunotherapy. A Phase II trial investigated the synergistic effects of tislelizumab (a modified anti-PD-1 mAb designed to reduce the interaction with FcγR on the macrophages to prevent antibody-dependent phagocytosis) in conjunction with cetuximab and irinotecan. This study assessed the efficacy and safety of this combination in treating refractory *RAS*wt mCRC patients. Among the 33 patients evaluated, partial remission was achieved in 36% and stable disease in 42%, with a confirmed ORR of 33%. Manageable side effects, including rash, fatigue, and gastrointestinal disturbances, were observed [97].

In CRC, natural killer (NK) cells in patients exhibit abnormal behavior and reduced function compared with those from healthy individuals. Combining cetuximab with cytokine therapies, such as IL-2 and IL-15, is considered to potentially enhance the action of cetuximab through the improved function and activity of NK cells [98,99]. SOT101, a fusion protein of IL-15 receptor α and IL-15, has shown promise in stimulating both subsets of NK cells and enhancing their cytotoxicity [100]. Preclinical studies and initial findings from the clinical study SC103 suggested that SOT101 could increase the activity of cetuximab by augmenting NK cell-mediated ADCC [98]. The SC103 clinical study (AURELIO-03; NCT04234113) investigated the safety and preliminary efficacy of SOT101 as a monotherapy and combined with pembrolizumab in advanced/metastatic solid tumor patients. Two mCRC patients were enrolled in the combination part, with one patient achieving stable disease [101]. The Phase II SC105 trial (AURELIO-05; NCT05619172) is ongoing, testing the combination of SOT101 with cetuximab in patients with *RAS*wt CRC.

## 5. Other Strategies with Cetuximab Combinations

### 5.1. Antiangiogenics and Cetuximab

Dual-antibody therapy combining bevacizumab and an anti-EGFR mAb is not recommended for the treatment of mCRC patients. While the Phase II BOND-2 trial reported promising results in terms of the response rates and survival, it was a small study conducted in specialized centers, limiting its applicability to a broader patient population [102]. The CAIRO2 and PACCE studies showed that dual-antibody therapy (combining bevacizumab with cetuximab and panitumumab, respectively) had a potentially detrimental effect and increased toxicity in the context of first-line therapy [103,104]. In the PACCE trial, a Phase III study investigating the combination of chemotherapy and bevacizumab with or without panitumumab, the patients were randomly assigned to receive FOLFOX or FOLFIRI according to the investigator’s discretion. The addition of panitumumab to these regimens led to a reduction in PFS for both wt and mutant *KRAS* tumor patients compared with those receiving FOLFOX or FOLFIRI with bevacizumab alone. Similarly, the CAIRO2 trial explored the combination of capecitabine, oxaliplatin, and bevacizumab, with or without cetuximab. Among the overall patient population, the inclusion of cetuximab resulted in a statistically significant decrease in the median PFS and an increase in Grade 3/4 adverse events. Notably, in patients with wild-type *KRAS*, there was no significant change in the median PFS with the addition of cetuximab. However, in those with mutant KRAS tumors, the use of cetuximab with chemotherapy and bevacizumab led to a statistically significant decrease in the median PFS compared with those who were not treated with a cetuximab-containing regimen. Combined, the findings from the PACCE and CAIRO2 trials highlighted that incorporating EGFR inhibitors into the first-line treatment for metastatic colorectal cancer (mCRC) does not improve the clinical outcomes, regardless of the KRAS mutational status, and may potentially diminish the overall effectiveness.

The possible benefits of combining cetuximab and ramucirumab were explored in the E7208 trial, but preliminary results showed a similar PFS with additional toxicity, indicating that further investigations are needed for the appropriate patient subgroups [105].

### 5.2. PI3KCA and MET

The phosphatidylinositol 3-kinase (PI3K) pathway plays a significant role in apoptosis, cellular proliferation, and motility. Mutations in *PI3KCA* are prevalent in mCRC patients, exhibiting a prevalence of 10–20%, mostly in the Exon 9 and Exon 20 hot spots of *PIK3CA*. The presence of *PIK3CA* mutations alongside *RAS*wt in mCRC patients negatively impacted the responses to anti-EGFR treatments, highlighting the importance of considering treatment strategies. Studies have indicated that *PIK3CA* mutations can predict resistance to EGFR inhibitors in CRC patients, emphasizing the need for personalized treatment approaches [106,107]. Moreover, mutations in *PIK3CA* were detected in the ctDNA of mCRC patients who had been treated with cetuximab, suggesting they contribute to acquiring cetuximab resistance [108].

While PI3K inhibition alone has shown limited clinical benefit, investigations into PI3Kα-inhibitor-containing regimens are ongoing [109,110]. Early clinical trials are studying the activity of diverse oral selective PI3K inhibitors in refractory mCRC patients: MEN1611 combined with cetuximab is being studied in patients with ctDNA that is positive for *PI3K*a mutations (NCT04495621); and inavolisib is being tested in a Phase I umbrella trial, in combination with cetuximab or bevacizumab, depending on the *RAS* status, in ctDNA after progression on an EGFR inhibitor (NCT04929223).

METamp is a secondary genetic alteration observed in mCRC patients with acquired resistance to anti-EGFR therapy, potentially driving the progression of the disease [111]. Combining the inhibition of MET, particularly with a highly selective MET tyrosine kinase inhibitor such as tepotinib, along with an anti-EGFR agent offers a promising strategy to control the disease. A Phase II study is being conducted to explore the potential of combining tepotinib with cetuximab in patients with RAS/*BRAF* wild-type left-sided mCRC who have acquired resistance to anti-EGFR antibody therapy due to METamp (NCT04515394) [112].

### 5.3. Other Combinations

In a Phase II/III trial, the potential of dalotuzumab (an anti-insulin-like growth factor-1 receptor monoclonal antibody) was investigated in combination with the standard therapy for chemorefractory *KRAS* wt mCRC patients. Patients were assigned to receive different doses of dalotuzumab or a placebo alongside cetuximab and irinotecan. The study was terminated early due to futility, with no significant improvements in the PFS or OS in the dalotuzumab arms [113].

## 6. Conclusions

In conclusion, cetuximab has emerged as a crucial and versatile player in the personalized targeted therapy approach for mCRC patients. It serves as an effective inhibitor of the EGFR signaling pathway, enhancing targeted therapy in molecularly selected mCRC populations. Cetuximab shows potential when combined with other targeted therapies for patients with tumors harboring the *BRAF* V600E and *KRAS* G12C mutations, expanding the treatment options for these subsets of mCRC patients. Additionally, the integration of immunotherapy with combination therapy with cetuximab presents an exciting avenue for further exploration, with ongoing trials investigating various regimens. These findings underscore the complexity of optimizing the management of mCRC and highlight the need for personalized approaches. Further research and clinical trials will continue to refine the role of cetuximab in the evolving landscape of mCRC therapy.

## Figures and Tables

**Figure 1 cancers-16-00412-f001:**
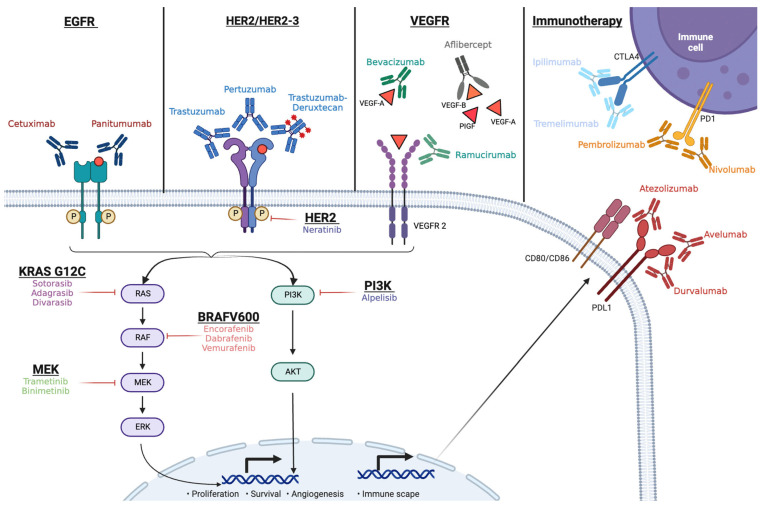
MAPK pathway and immunotherapy, identifying key inhibitory sites for a comprehensive therapeutic intervention.

**Table 1 cancers-16-00412-t001:** Key completed and ongoing trials evaluating *KRAS* G12C inhibitors in combination with EGFR inhibitors.

Clinical Trial	Phase	Treatment	Population	Outcome(s)
Completed/ongoing trial (with published data)				
CodeBreaK 101 (NCT04185883)	I	Sotorasib + panitumumab	Refractory mCRC	PFS: 5.7 months, ORR: 30% (12/40)
KRYSTAL-01 (NCT03785249)	I	Adagrasib + cetuximab	Refractory mCRC	PFS: 6.9 months, OS: 13.4 months ORR: 46% (13/28) PFS: 6.9 months
NCT04449874	I	Divarasib + cetuximab	Refractory mCRC	ORR: 62% (18/29)
CodeBreak 300 (NCT05198934)	III	Sotorasib + panitumumab versus investigator’s choice	mCRC with ≥1 prior line of therapy	PFS: 5.6 months with 960 mg sotorasib; 3.9 months with 240 mg sotorasib
Ongoing trials (data not yet available)				
KRYSTAL-10 (NCT04793958)	III	Adragasib + cetuximab versus FOLFOX/FOLFIRI	Refractory mCRC	OS; PFS
NCT05002270	I/II	JAB-21822 +/− cetuximab	Refractory mCRC	Safety; ORR; DOR

DOR: duration of response; ORR: overall response rate; OS: overall survival; PFS: progression-free survival.

**Table 2 cancers-16-00412-t002:** Summary of the key ongoing and published clinical trials investigating cetuximab combined with immunotherapy.

Clinical Trial	Phase	Treatment	Population	Outcomes
AVETUXIRI NCT03608046	II	Cetuximab + avelumab	MSS refractory mCRC A: *RASwt*/B: *RASmut*	ORR: 30%; 6 mo PFS: 40.0%
CAVE study NCT04561336	II	Cetuximab + avelumab	*RASwt* mCRC response to first-line anti-EGFR ≥ third-line	ORR: 7%; mOS: 11.6 mo; mPFS: 3.6 mo
AVETUX NCT03174405	II	Avelumab + cetuximab + FOLFOX	*RAS/BRAF*wt mCRC, first-line	ORR: 79.5%; DCR: 92.3%
NCT02713373	Ib/II	Pembrolizumab + cetuximab	MSS/MSI RAS/*BRAF*wt mCRC patients eligible for anti-EGFR therapy	ORR: 2.6%; 6 mo PFS: 31%
CAVE II trial NCT05291156	II	Cetuximab vs. cetuximab + avelumab	Response of *RASwt* mCRC to first-line anti-EGFR ≥ third-line	Ongoing trial
NCT05167409	II	Evorpacept (ALX148) + cetuximab + pembrolizumab	MSS refractory	Ongoing trial

mo: months; mOS: median overall survival; MSS: microsatellite stable; ORR: overall response rate; mPFS: median progression-free survival; wt: wild-type.

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
