# Peer review of "Cetuximab as a Key Partner in Personalized Targeted Therapy for Metastatic Colorectal Cancer"

_cancers, 2024, doi:10.3390/cancers16020412_

Round 1

Reviewer 1 Report

Comments and Suggestions for Authors

The review by Gonzalez and colleagues deals with current perspectives on treatment strategies involving cetuximab for metastatic colorectal cancer.

As a general comment, many of the evidences gleaned by the Authors rightly include studies with panitumumab. However, since the topic of this review is focused on cetuximab, I suggest to describe in paragraph 2 (Mechanism of action) the existing differences between these two antibodies. Some sections report on adverse events, other not. If safety has to be reported, this need to be more detailed and consistent throughout the manuscript. While some phase I studies include treatment arms with cetuximab, the rationale behind needs to be addressed: for clarity sake, these studies should not be reported in tables.

Specific comments:

1) The introduction needs to be streamlined

2) Figure 1 needs to be more detailed, particularly on the immunotherapy side. For clarity sake, all agents currently approved by regulatory agencies have to be reported

3) The content of lines 70-82 is broadly overlapping with lines 110-122.

4) Since this review deals with cetuximab, lines 149-158 should report on rechallenge studies with cetuximab (Cricket, Cave, Jaccro) putting them into perspective

5) Table 1, Ongoing trial section, does not provide additional information to the reader.

6) While mentioning current studies on neratinib + cetuximab, the findings of the study by Kavuri et al (ancer discov 2015) should be acknowledged:

7) section regarding cetuximab and immunotherapy should focus on the role of ADCC in the rationale for combining cetuximab + IO.

8) Provide more information on Cairo2 and Pacce trials to highlight the reasons behind their negative results

Comments on the Quality of English Language

Good quality English

Author Response

Thank you very much for your valuable comments and the time you spent on the review. The changes have been introduced in red in the manuscript, and the deleted portions have been marked with strikethrough (example)

The review by Gonzalez and colleagues deals with current perspectives on treatment strategies involving cetuximab for metastatic colorectal cancer.

As a general comment, many of the evidences gleaned by the Authors rightly include studies with panitumumab. However, since the topic of this review is focused on cetuximab, I suggest to describe in paragraph 2 (Mechanism of action) the existing differences between these two antibodies. 

Thank you for your suggestion. We appreciate your feedback. In response to your comment, we have incorporated a dedicated paragraph in the "Mechanism of Action" section, elucidating the existing differences in mechanisms of action between cetuximab and panitumumab.

Some sections report on adverse events, other not. If safety has to be reported, this need to be more detailed and consistent throughout the manuscript. 

Thank you for your feedback. The primary focus of our review is not on toxicity; however, safety data have been included where relevant. We have reviewed and revised the sections reporting adverse events to ensure consistency and detail in line with the scope of our manuscript (eliminated page 8 line 246: (The escalation of dosage was carried out across four pre-defined…); page 9 line 388: (Of the 77 patients enrolled, 14% presented…)

While some phase I studies include treatment arms with cetuximab, the rationale behind needs to be addressed: for clarity sake, these studies should not be reported in tables.

Thank you for your suggestion. We have addressed the concern by removing the clinical trials NCT04449874 and NCT04956640 from Table 1. However, we have retained EC CodeBreaK 101 and KRYSTAL-01 in Table 1 due to their significant contribution to the pharmacological development of KRASG12C inhibitors. In Table 2, we have excluded AURELIO-03 (NCT04234113) as per your recommendation. We believe these adjustments improve the clarity and relevance of the information presented. If you have any further comments or concerns, please feel free to share them. 

Specific comments:

1) The introduction needs to be streamlined

Thank you for your suggestion. We have revised and eliminated some parts of the introduction to enhance its streamlined nature.

2) Figure 1 needs to be more detailed, particularly on the immunotherapy side. For clarity sake, all agents currently approved by regulatory agencies have to be reported

Thank you for your feedback. We appreciate your suggestion regarding Figure 1. While we acknowledge the importance of providing comprehensive information, the primary objective of the figure is to offer a broad overview rather than an exhaustive list of all approved agents and their mechanisms of action in targeted therapy and immunotherapy.

3) The content of lines 70-82 is broadly overlapping with lines 110-122.

Thank you for bringing this to our attention. We have addressed the issue by summarizing the content of lines 70-82 to reduce the overlap with lines 110-122.

4) Since this review deals with cetuximab, lines 149-158 should report on rechallenge studies with cetuximab (Cricket, Cave, Jaccro) putting them into perspective

Thank you very much for your input. We have made modifications to lines 149-158 to provide a clearer perspective on rechallenge studies of rechallenge and cetuximab,. Additionally, I would like to note that the CAVE study is specifically mentioned in the "Immunotherapy and cetuximab" section

5) Table 1, Ongoing trial section, does not provide additional information to the reader.

Thank you for your feedback. We have retained the KRYSTAL-10 trial in Table 1, as it is a phase III clinical trial that we consider important for the pharmacological development and positioning of these targeted treatments. If you have any further suggestions or concerns, we are open to additional feedback.

6) While mentioning current studies on neratinib + cetuximab, the findings of the study by Kavuri et al (ancer discov 2015) should be acknowledged:

Thank you for bringing up the study by Kavuri et al. We appreciate the valuable input, and we have now included a section acknowledging the findings of the mentioned study in our revised manuscript (in the Page 8 in neratinib paraph).

7) section regarding cetuximab and immunotherapy should focus on the role of ADCC in the rationale for combining cetuximab + IO.

Thank you for your feedback. We have addressed your suggestion by adding clarifications in the existing section, emphasizing the role of ADCC.

8) Provide more information on Cairo2 and Pacce trials to highlight the reasons behind their negative results

Thank you for your suggestion. We have added clarifications in the existing section explaining the CAIRO2 and PACCE trials.

Reviewer 2 Report

Comments and Suggestions for Authors

This review summarizes the results of monotherapy and combination therapy for colorectal cancer using cetuximab. The review is quite interesting and complete, and also corresponds to the profile of the journal. The author's approach to systematizing the results deserves attention. However, there are recommendations:

1) In the introduction, it should be noted how this review compares favorably with other reviews, summing up the references to them.

2) Some recent reviews on chemotherapy for colorectal cancer were not the focus of the manuscript. For example https://doi.org/10.3390/ma14092440.

3) The time period for which the literature is reviewed should be outlined.

4) The potential of nanomedicine to deliver cetuximab has not been shown.

These recommendations could improve the review, which is worthy of publication after corrections.

Author Response

This review summarizes the results of monotherapy and combination therapy for colorectal cancer using cetuximab. The review is quite interesting and complete, and also corresponds to the profile of the journal. The author's approach to systematizing the results deserves attention. However, there are recommendations:

Thank you for your insightful feedback. We appreciate your positive evaluation of our review.

1) In the introduction, it should be noted how this review compares favorably with other reviews, summing up the references to them.

In response to your recommendation to highlight how our review compares favorably with others in the introduction, we have duly noted this suggestion. In the final paragraph, there is a statement addressing this aspect, aiming to articulate the distinctive contributions of our review in comparison to existing literature.

2) Some recent reviews on chemotherapy for colorectal cancer were not the focus of the manuscript. For example https://doi.org/10.3390/ma14092440.

Thank you for bringing this to our attention. We appreciate your understanding that our focus in a review is not to extensively reference other reviews within the manuscript. This decision aligns with our intention to maintain the integrity and originality of our review.

3) The time period for which the literature is reviewed should be outlined.

Thank you for your observation. We appreciate your recommendation regarding the clarification of the time period covered in the literature review. Our manuscript is not intended to be a systematic review or meta-analysis, that's why we have chosen to avoid this to avoid confusion. 

4) The potential of nanomedicine to deliver cetuximab has not been shown.

Thank you for addressing the potential of nanomedicine in delivering cetuximab. We have indeed focused on reviewing and summarizing primarily clinical evidence in our manuscript. Consequently, these technologies have not been integrated into the review due to their current status as preclinical evidence, at least to the extent of our knowledge.

These recommendations could improve the review, which is worthy of publication after corrections.

Round 2

Reviewer 1 Report

Comments and Suggestions for Authors

The revised version of the manuscript is improved. However, I have still some concerns on figure 1. While the scope of figure 1 is to provide a broad overview on targeted agents and antibodies against key immuno-oncology targets, it is not clear why some agents have been mentioned and others not. For instance, the KRAS inhibitors list appears to be exceedingly detailed (with agents under study even in FIH trial), whereas the IO section includes just a few antibodies, thereby making the content of figure 1 poorly informative and unbalanced. I therefore suggest to modify figure 1.

Comments on the Quality of English Language

Good quality English. 

Author Response

Dear Reviewer,

Thank you for your valuable feedback. We appreciate your thoughtful evaluation of the revised manuscript.

Regarding Figure 1, we have carefully considered your concerns about the balance and informativeness of the content.

In response to your comments, we have revised the figure to address the imbalance you noted. We have streamlined the list of KRAS inhibitors, and have expanded the immunotherapy section to include a more comprehensive representation of antibodies against key targets. Additionally, we have incorporated the information about the neratinib. We have also changed the title of the figure.

We hope that these modifications enhance the overall clarity and balance of Figure 1.

Thank you for your and Best Regards,

Reviewer 2 Report

Comments and Suggestions for Authors

Although I believe that taking into account previous literature reviews in no way casts a shadow on the originality of your review and is useful from the perspective of the breadth of literature cited, I do not mind if the manuscript is published.

Author Response

Dear collegues,
Thank you